# Calibrating Neural Simulation-Based Inference with Differentiable Coverage Probability

**Maciej Falkiewicz**[1,2]    **Naoya Takeishi**[3,4]    **Imahn Shekhzadeh**[1,2]    **Antoine Wehenkel**[5]
**Arnaud Delaunoy**[5]    **Gilles Louppe**[5]    **Alexandros Kalousis**[2]
[1]Computer Science Department, University of Geneva    [2]HES-SO/HEG Genève
[3]The University of Tokyo    [4]RIKEN    [5]University of Liège
{maciej.falkiewicz, imahn.shekhzadeh, alexandros.kalousis}@hesge.ch
ntake@g.ecc.u-tokyo.ac.jp
{antoine.wehenkel, a.delaunoy, g.louppe}@uliege.be

## Abstract

Bayesian inference allows expressing the uncertainty of posterior belief under a probabilistic model given prior information and the likelihood of the evidence. Predominantly, the likelihood function is only implicitly established by a simulator posing the need for simulation-based inference (SBI). However, the existing algorithms can yield overconfident posteriors (Hermans *et al.*, 2022) defeating the whole purpose of credibility if the uncertainty quantification is inaccurate. We propose to include a calibration term directly into the training objective of the neural model in selected amortized SBI techniques. By introducing a relaxation of the classical formulation of calibration error we enable end-to-end backpropagation. The proposed method is not tied to any particular neural model and brings moderate computational overhead compared to the profits it introduces. It is directly applicable to existing computational pipelines allowing reliable black-box posterior inference. We empirically show on six benchmark problems that the proposed method achieves competitive or better results in terms of coverage and expected posterior density than the previously existing approaches.

## 1   Introduction

Inverse problems [46] appear in a variety of fields of human activity, including engineering [2], geology [8], medical imaging [29], economics [21], and particle physics [1]. In some domains, uncertainty quantification is required for results to be used in the real-world [51]. A particularly well-suited tool for this task is Bayesian inference, where the procedure of updating prior beliefs (knowledge) with observed evidence is formally described in the framework of probability theory. In order to compute the posterior probability according to Bayes' rule, the likelihood function has to be evaluated. Often, it is only implicitly established by a simulator which given input parameters generates (possibly stochastically) outputs. Such a situation requires the use of likelihood-free techniques, also referred to as simulation-based inference (SBI) [9].

There exist two major classical approaches that address the problem of Bayesian inference in the likelihood-free setting. In the first one, simulated data are used to estimate the distribution of expert-crafted low-dimensional summary statistics with a nonparametric method like kernel density estimation [14]. Next, the approximate density is used as a likelihood surrogate turning the problem into likelihood-based inference. In the second approach, termed as Approximate Bayesian Computation (ABC) [40, 38, 48, 42, 3, 44] no data is simulated upfront, but instead for every newly appearing observation (instance of the inverse problem), an independent sampling-rejection procedure with a simulator in-the-loop is run. Given some proposal distribution, candidate

parameters are accepted if the simulator's output is in sufficient agreement with the observation. While ABC effectively sorts out the lack of explicit likelihood, it is generally costly, especially when the underlying simulator is computationally expensive.

Deep Neural Networks have been increasingly utilized in SBI to alleviate these costs, by acting as universal and powerful density estimators or surrogating other functions. The following approaches of neural SBI can be identified: Neural Likelihood Estimation (NLE) [35], where the missing likelihood function is approximated based on the simulator; Neural Ratio Estimation (NRE) [19, 15, 32], where the likelihood-to-evidence ratio (other ratios have also been considered) is approximated; Neural Posterior Estimation (NPE) [34, 30, 18], where the learned density estimator directly solves the inverse problem for a given observation. In the last category the score-based methods [41, 16] are emerging, where the posterior is implicitly expressed by the learned score function. All of the approaches mentioned above amortize the use of a simulator, however, inference with NLE and NRE still requires the use of Markov chain Monte Carlo [7], while NPE can be fully amortized with the use of appropriate generative models [47].

The question of reliability is raised with the use of approximate Bayesian inference methods. Many works have been devoted to inference evaluation techniques in the absence of the ground-truth posterior reference [45, 53, 27, 28]. Lueckmann et al. [31] propose an SBI benchmarking protocol taking into account the effect of the simulation budget, with the observation that the assessment's conclusions will strongly depend on the metric chosen. Hermans et al. [20] suggest relying on coverage analysis of posterior's multi-dimensional credible regions, showing that numerous SBI techniques yield overconfident posteriors. In a follow-up work [11] they introduce a balancing regularizer for training NRE, and show empirically that it leads to the avoidance of overconfident solutions. Very recently, an interesting study by Delaunoy et al. [12] extends the applicability of balancing regularizer to the NPE approach. An alternative perspective is studied in Dey et al. [13] where an additional regression model for posthoc re-calibration based on Probability Integral Transform is learned for the outputs of a pre-trained inference model.

In this work, we introduce and empirically evaluate a new regularizer that directly targets the coverage of the approximate posterior. It is applicable to NRE and NPE based on neural density estimators such as Normalizing Flows [26, 36]. As an additive term, it is easy for practitioners to use as a plug-in to existing pipelines. The additional computational overhead introduced by the regularizer is justified in view of the advantages it brings.

The rest of the work is structured as follows. Section 2 introduces the formal description of the studied problem and presents the coverage analysis that is the motivation for the presented method. Section 3 describes the proposed method and a differentiable formulation of the regularizer that allows gradient-based learning. Section 4 shows the empirical results of applying the method on a wide range of benchmark problems, as well as providing a hyper-parameter sensitivity analysis. In section 5, we provide summarize our work with conclusions and potential future directions.

## 2   Background

In this work, we consider the problem of estimating posterior distribution $p(\theta|x)$ of model parameters $\theta$, given the observation $x$. We approach the problem from a likelihood-free [43] perspective, where the model's likelihood $p(x|\theta)$ cannot be evaluated. Instead, there exists a simulator that generates observations $x$ based on the input parameters $\theta$. We can prepare a dataset $\{(\theta_i, x_i)\}_{i=1}^N$, by sampling parameters from a prior $p(\theta)$ and running them through the simulator. In that way, the problem of finding the approximate posterior marked as $\hat{p}(\theta|x)$ turns into a supervised learning problem. However, it is important to note that usually there is not only a single generating parameter $\theta$ for a given $x$ possible. Instead, for each observation multiple parameter values are plausible [23]. These values form a posterior distribution $p(\theta|x)$ under a prior distribution $p(\theta)$.

Assessing the performance of an SBI technique is challenging due to the non-availability of the ground-truth posterior $p(\theta|x)$ in the general case. Theoretically, methods such as the rejection-sampling ABC [44] are proven to converge to $p(\theta|x)$ under an infinite simulation budget but often require a large number of simulation runs to achieve a good approximation. Avoiding this costly procedure is exactly what motivates the introduction of different techniques. Therefore, it is common to resort to performance metrics evaluated on test datasets sampled from the simulator.

If a Bayesian inference method allows $\hat{p}(\theta|x)$ evaluation, its performance can be assessed by looking at the expected value of the approximate log posterior density of the data-generating parameters (nominal parameters) $\mathbb{E}_{p(\theta,x)}[\log \hat{p}(\theta|x)]$, estimated over a test dataset. In domains such as natural sciences, it is also accurate uncertainty quantification that matters, i.e. reliable posterior density estimation given the available evidence in data. It has been shown in Hermans et al. [20] that approaches yielding high expected posterior density values can give overconfident posteriors, and therefore, verifying the quality of credible regions returned by inference engines as well as designing more reliable ones is essential. In this work, we define a regularizer suitable for training NRE and NPE based on the Simulation-Based Calibration (SBC) diagnostic of Talts et al. [45]. To make the paper self-contained, in the rest of Section 2 we will introduce SBC formally, adapting the definitions and lemmas from Lemos et al. [27], and making the link with the coverage analysis from Hermans et al. [20].

## 2.1 Coverage analysis

Coverage analysis offers a way to evaluate the quality of an approximate posterior. Given an observation and an estimated distribution, one can assess the probability that a certain credible region [22] of the parameter space contains the nominal parameter value found in the dataset. For a conditional probability distribution, a credible region at level $1 - \alpha$ is a region of the support to which the parameter used to generate the conditioning observation belongs with a $1 - \alpha$ probability, called the credibility level. There are infinitely many credible regions at any given level, and coverage analysis requires a particular way of choosing one. In this work, we will consider the Highest Posterior Density Region (HPDR).

**Definition 1** (Highest Posterior Density Region)**.**

$$\Theta_{\hat{p}(\theta|x)}^{HPDR}(1 - \alpha) := \arg\min_{\Theta} |\Theta| \;\; s.t. \;\; \int_{\theta \in \Theta} \hat{p}(\theta|x)\mathrm{d}\theta = 1 - \alpha, \tag{1}$$

*where* $|\Theta| := \int_{\Theta} \mathrm{d}\mu$ *is the volume of a subset* $\Theta$ *of the parameters' space, under some measure* $\mu$.

Next, we introduce the central concept of coverage analysis.

**Definition 2** (Expected Coverage Probability)**.** *The* Expected Coverage Probability (ECP) *over a joint distribution* $p(\theta, x)$ *for a given* $\Theta_{\hat{p}(\theta|x)}^{HPDR}(1 - \alpha)$ *is*

$$ECP\left(\Theta_{\hat{p}(\theta|x)}^{HPDR}(1 - \alpha)\right) := \mathbb{E}_{p(\theta,x)}\left[\mathbb{1}\left[\theta \in \Theta_{\hat{p}(\theta|x)}^{HPDR}(1 - \alpha)\right]\right], \tag{2}$$

*where* $\mathbb{1}[\cdot]$ *is the indicator function.*

While the expectation in eq. (2) can be easily estimated by using a set of i.i.d. samples from the joint distribution $p(\theta, x)$, the challenge of finding $\Theta_{\hat{p}(\theta|x)}^{\mathrm{HPDR}}(1 - \alpha)$ remains non-trivial. In general, directly following eq. (1) requires solving an optimization problem. This would be prohibitive to do for every instance during training, thus it is crucial to benefit from the two lemmas we introduce below.

**Lemma 1.** *A pair* $(\theta^*, x^*)$ *and a distribution* $\hat{p}(\theta|x)$ *uniquely define an HPDR:*

$$\Theta_{\hat{p}(\theta|x)}^{HPDR}(1 - \alpha_{HPDR}(\hat{p}, \theta^*, x^*)) := \{\theta \mid \hat{p}(\theta|x^*) \geqslant \hat{p}(\theta^*|x^*)\}, \tag{3}$$

*where*

$$\alpha_{HPDR}(\hat{p}, \theta^*, x^*) := \int \hat{p}(\theta|x)\mathbb{1}[\hat{p}(\theta|x^*) < \hat{p}(\theta^*|x^*)]d\theta. \tag{4}$$

See proof in Appendix A .

**Lemma 2.**

$$ECP\left(\Theta_{\hat{p}(\theta|x)}^{HPDR}(1 - \alpha)\right) = \mathbb{E}_{p(\theta,x)}\left[\mathbb{1}[\alpha_{HPDR}(\hat{p}, \theta, x) \geqslant \alpha]\right] \tag{5}$$

See proof of Lemma 1 in Lemos et al. [27].

If $ECP\left(\Theta_{\hat{p}(\theta|x)}^{\mathrm{HPDR}}(1 - \alpha)\right)$ is equal to $1 - \alpha$ the inference engine yields posterior distributions that are calibrated at level $1 - \alpha$, or simply calibrated if the condition is satisfied for all $\alpha$. The ground-truth posterior $p(\theta|x)$ is calibrated, but another such distribution is the prior $p(\theta)$ as discussed in Hermans

et al. [20] and Lemos et al. [27]. Therefore, coverage analysis should be applied in conjunction with the standard posterior predictive check. Formally, we aim for:

$$p^*(\theta|x) := \underset{\hat{p}(\theta|x)}{\arg\max} \, \mathbb{E}_{p(\theta,x)}\left[\log \hat{p}(\theta|x)\right] \quad \text{s.t.} \quad \forall \alpha \, \text{ECP}\left(\Theta_{\hat{p}(\theta|x)}^{\text{HPDR}}(1-\alpha)\right) = 1 - \alpha \qquad (6)$$

In Hermans et al. [20] it has been pointed out that for practical applications often the requirement of the equality sign in the condition in eq. (6) can be relaxed and replaced with $\text{ECP}\left(\Theta_{\hat{p}(\theta|x)}^{\text{HPDR}}(1-\alpha)\right)$ above or equal the $1 - \alpha$ level. Such a posterior distribution is called conservative.

We define the *calibration error* as

$$\frac{1}{M} \sum_{i=1}^{M} |(1 - \alpha_i) - \text{ECP}\left(\Theta_{\hat{p}(\theta|x)}^{\text{HPDR}}(1-\alpha_i)\right)|, \qquad (7)$$

and the *conservativeness error* as

$$\frac{1}{M} \sum_{i=1}^{M} \max\left((1 - \alpha_i) - \text{ECP}\left(\Theta_{\hat{p}(\theta|x)}^{\text{HPDR}}(1-\alpha_i)\right), 0\right), \qquad (8)$$

where $\{\alpha_i\}_{i=1}^{M}$ is a set of credibility levels of arbitrary cardinality such that $\forall_i \, 0 < \alpha_i < 1$; we can exclude $\alpha = 0$ and $\alpha = 1$ because calibration/conservativeness at these levels is achieved by construction. For a motivating graphical example of coverage analysis of an overconfident distribution see appendix D.

## 3 Method

We propose a method supporting the training of neural models underlying SBI engines such that the resulting distributions are calibrated or conservative against the real $p(\theta, x)$ in the data. In addition to the standard loss term in the optimization objective function, we include a regularizer corresponding to differentiable relaxations of the calibration (conservativeness) error as defined in eq. (7) (eq. (8)). In the remainder of this section, we will show how to efficiently estimate the expected coverage probability during training and how to alleviate the arising non-continuities to allow gradient-based optimization. After a few simplifications, the regularizer will become a mean square error between the sorted list of estimated $\alpha_{\text{HPDR}}(\hat{p}, \theta, x)$s and a step function on the $(0, 1)$ interval.

The proposed method is based on the Monte Carlo (MC) estimate of the integral in eq. (4):

$$\hat{\alpha}_{\text{HPDR}}^{L}(\hat{p}, \theta^*, x^*) := \frac{1}{L} \sum_{j=1}^{L} \mathbb{1}\left[\hat{p}(\theta_j|x^*) < \hat{p}(\theta^*|x^*)\right], \qquad (9)$$

where $\theta_1, ..., \theta_L$ are samples drawn from $\hat{p}(\theta|x)$. The entity defined in eq. (9) is a special case of what is called *rank statistic* in Talts et al. [45] (with the $1/L$ normalization constant skipped there). The non-normalized variant is proven to be uniformly distributed over the integers $[0, L]$ for $\hat{p}(\theta|x) = p(\theta|x) \, \forall(\theta, x) \sim p(\theta, x)$, which is the *accurate posterior* as defined in Lemos et al. [27]. Taking $L \to \infty$ and applying the normalization (as done in eq. (9)) one shows that distribution of $\hat{\alpha}_{\text{HPDR}}^{\infty}(\hat{p}, \theta^*, x^*)$ converges to a standard continuous uniform distribution (proof in appendix A.2).

We return to coverage analysis with the following remark:

**Remark 1.** *An inference engine yields calibrated posterior distributions if and only if it is characterized by uniformly distributed $\alpha_{HPDR}(\hat{p}, \theta, x)$s.*

See proof in Appendix A .

Following remark 1, we will enforce the uniformity of the normalized rank statistics which coincides with a necessary, though not sufficient, condition for the approximate posterior to be accurate. The one-sample Kolmogorov–Smirnov test provides a way to check whether a particular sample set follows some reference distribution. We will use its test statistic to check, and eventually enforce, the condition. An alternative learning objective is to directly minimize the calibration (conservativeness) error at arbitrary levels with ECP computed following eq. (5).

## 3.1 One-sample Kolmogorov–Smirnov test

Let $B$ be a random variable taking values $\beta \in \mathcal{B}$. Given a sample set $\hat{\mathcal{B}} = \{\beta_i | i = 1, \ldots, N\}$, the one-sample Kolmogorov–Smirnov test statistic [5] for r.v. $B$ is defined as:

$$D_N = \sup_{\beta \in \mathcal{B}} |F_N(\beta) - F(\beta)|, \tag{10}$$

where $F_N$ is the empirical cumulative distribution function (ECDF) of the r.v. $B$, with regard to the sample set $\hat{\mathcal{B}}$ [10]. $F$ is the reference cumulative distribution function (CDF) against which we compare the ECDF.

In the proposed method, we instantiate the KS test statistic for r.v. $\alpha_{\text{HPDR}}(\hat{p}, \theta, x)$, where $(\theta, x) \sim p(\theta, x)$, thus $F_N$ is its ECDF, which we estimate from a sample set $\hat{\mathcal{A}} := \{\hat{\alpha}^L_{\text{HPDR}}(\hat{p}, \theta_i, x_i) | (x_i, \theta_i) \sim p(x, \theta), i = 1, \ldots, N\}$. The reference CDF, $F$, is that of the standard uniform distribution because we want to check and enforce the uniformity of the elements of $\hat{\mathcal{A}}$.

By the Glivenko–Cantelli theorem [49] we know that $D_N \to 0$ almost surely with $N \to \infty$ if samples follow the $F$ distribution. The claim in the opposite direction, that is if $D_N \to 0$ then samples follow $F$, is also true and we show that in appendix A.3. As a result, minimizing $D_N$ will enforce uniformity and thus posterior's calibration. We can upper-bound the supremum in eq. (10) with the sum of absolute errors (SAE), but instead, to stabilize gradient descent, we use the mean of squared errors (MSE), which shares its optimum with SAE. Minimizing the one-sample KS test statistic is just one of the many ways of enforcing uniformity. We can deploy any divergence measure between the sample set and the uniform distribution.

## 3.2 Approximating $D_N$

In practice we cannot evaluate the squared errors over the complete support of the r.v., but only over a finite subset. There are two ways of evaluating this approximation which differ by how we instantiate $F_N(\alpha)$ and $F(\alpha)$.

**Direct computation** In the first, we compute $F_N(\alpha) = \sum_i \mathbb{1}[\alpha_i \leq \alpha]/N$ summing over all $\alpha_i \in \hat{\mathcal{A}}$ and use the fact that $F(\alpha) = \alpha$ for the target distribution. To approximate $D_N$ we aggregate the squared error over an arbitrary set of $\alpha_k$ values: $\sum_k (F_N(\alpha_k) - \alpha_k)^2$. Including this term as a regularizer means that we need to backpropagate through $F_N(\alpha_k)$. Such an approach can be seen as directly aiming for calibration at arbitrary levels $\alpha_k$, i.e. applying eq. (5) to evaluate the calibration error.

**Sorting-based computation** Alternatively, we can sort the values in $\hat{\mathcal{A}}$ to get the ordered sequence $(\alpha_i | i := 1 \ldots N)$. Given the sorted values, $\alpha_i$, and their indices, $i$, the $F_N(\alpha_i)$ values are given directly by the indices normalized by the sample size $(i/N)$, and by the corresponding target distribution $F(\alpha_i) = \alpha_i$, which leads to the $\sum_i (i/N - \alpha_i)^2$ approximation. Under this approach, the gradient will now flow through $F(\alpha_i)$. This is the approach that we use in our experiments.

## 3.3 Importance Sampling integration

In the definition of $\alpha_{\text{HPDR}}(\hat{p}, \theta^*, x^*)$ in eq. (4) as well as in its MC estimate $\hat{\alpha}^L_{\text{HPDR}}(\hat{p}, \theta^*, x^*)$ defined in eq. (9) we sample $\theta_j$s from the approximate posterior $\hat{p}(\theta|x)$. Optimizing the model constituting the approximate posterior requires backpropagating through the sampling operation which can be done with the reparametrization trick [24]. However, the reparametrization trick is not always applicable; for example, in NRE an MC procedure is required to sample from the posterior [19]. Here we can use instead self-normalized Importance Sampling (IS) to estimate $\hat{\alpha}^L_{\text{HPDR}}(\hat{p}, \theta, x)$ and lift the requirement of sampling from the approximate posterior. We will be sampling from a known proposal distribution and only evaluate the approximate posterior density. We thus propose to use IS integration [e.g., 17] to estimate the integral in eq. (4), in the following way:

$$\hat{\alpha}^{L,IS}_{\text{HPDR}}(\hat{p}, \theta^*, x^*) = \frac{\sum_{j=1}^L \hat{p}(\theta_j | x^*)/I(\theta_j) \mathbb{1}[\hat{p}(\theta_j | x^*) < \hat{p}(\theta^* | x^*)]}{\sum_{j=1}^L \hat{p}(\theta_j | x^*)/I(\theta_j)}. \tag{11}$$

**Algorithm 1** Computing the regularizer loss with calibration objective.

---

**Require:** Data batch $\{(\theta_i, x_i)\}_{i=1}^N$, model $\hat{p}(\theta|x)$, number of samples $L$, proposal distribution $I(\theta)$
**Ensure:** Regularizer's loss $R$
1: **for** $i \leftarrow 1$ to $N$ **do**
2: $\quad p_i \leftarrow \hat{p}(\theta_i|x_i)$
3: $\quad$ **for** $j \leftarrow 1$ to $L$ **do**
4: $\quad\quad \theta_i^j \sim I(\theta)$ $\qquad\qquad\qquad\qquad\qquad\qquad$ ▷ sampling from proposal distribution
5: $\quad\quad p_i^j \leftarrow \hat{p}(\theta_i^j|x_i)$
6: $\quad$ **end for**
7: $\quad \hat{\alpha}_{\text{HPDR}}^{L,IS}(\hat{p}, \theta_i, x_i) \leftarrow \frac{\sum_{j=1}^L p_i^j / I(\theta_i^j) \mathbb{1}\left[p_i^j < p_i\right]}{\sum_{j=1}^L p_i^j / I(\theta_i^j)}$ $\qquad\qquad\qquad$ ▷ eq. (11)
8: **end for**
9: $(\alpha_i | i = 1, \ldots, N) \leftarrow \text{sort}(\{\hat{\alpha}_{\text{HPDR}}^{L,IS}(\hat{p}, \theta_i, x_i) | i = 1, \ldots, N\})$
10: $R \leftarrow \frac{1}{N} \sum_i^N (i/N - \alpha_i)^2$ $\qquad\qquad\qquad\qquad\qquad$ ▷ the second term in eq. (12)

---

The $\theta_j$s are sampled from the proposal distribution $I(\theta)$. For a uniform proposal distribution, the $I(\theta_j)$ terms are constant and thus cancel out. By default, we suggest using the prior as the proposal distribution. While IS is not needed in the case of NPE, we use it also for this method to keep the regularizer's implementation consistent with the NRE setting.

### 3.4 Learning objective

So far we have shown how to derive a regularizer that enforces a uniform distribution of the normalized rank statistics that characterizes a calibrated model as shown in remark 1. This is also a necessary condition for the accuracy of the posterior as shown in Talts et al. [45] but not a sufficient one, as exemplified by the approximate posterior equal to the prior for which the regularizer trivially obtains its minimum value. Thus the final objective also contains the standard loss term and is a weighted sum of the two terms:

$$\frac{1}{N} \sum_i^N \log \hat{p}(\theta_i|x_i) + \lambda \frac{1}{N} \sum_i^N \left(\text{index}\left(\hat{\alpha}_{\text{HPDR}}^{L,IS}(\hat{p}, \theta_i, x_i)\right)/N - \hat{\alpha}_{\text{HPDR}}^{L,IS}(\hat{p}, \theta_i, x_i)\right)^2, \quad (12)$$

which we will minimize with AdamW optimizer. The first term in eq. (12) is the likelihood loss which we use for the NF-based NPE (or its corresponding cross-entropy loss for the NRE) and the second is the uniformity enforcing regularizer. The $\text{index}(\cdot)$ operator gives the index of its argument in the sequence of sorted elements of $\{\hat{\alpha}_{\text{HPDR}}^{L,IS}(\hat{p}, \theta_i, x_i) \mid i = 1, \ldots, N\}$. When regularizing for conservativeness we accept $\alpha_{\text{HPDR}}(\hat{p}, \theta^*, x^*)$s to first-order stochastically dominate [39] the target uniform distribution, thus the differences in the uniformity regularizer are additionally activated with a rectifier $\max(\cdot; 0)$ analogously to eq. (8). In appendix A.1 we sketch a proof of how minimizing such regularizer affects the learned model. The two main hyper-parameters of the objective function are the number of samples $L$ we use to evaluate eq. (11), and the weight of the regularizer $\lambda$ in eq. (12). In algorithm 1 we provide the pseudo-code to compute the uniformity regularizer.

To allow backpropagation through indicator functions, we use a Straight-Through Estimator [4] with the Hard Tanh [37] backward relaxation. In order to backpropagate through sorting operation we use its differentiable implementation from Blondel et al. [6].

## 4 Experiments

In our experiments[1], we basically follow the experimental protocol introduced in Hermans et al. [20] for evaluating SBI methods. We focus on two prevailing amortized neural inference methods, i.e. NRE approximating the likelihood-to-evidence ratio and NPE using conditional NF as the underlying model. In the case of NRE, there has been already a regularization method proposed [11] and we include it for comparison with our results, and mark it as BNRE. In the main text of the paper, we focus on the conservativeness objective. For the calibration objective, the results can be found in

---

[1]The code is available at `https://github.com/DMML-Geneva/calibrated-posterior`.

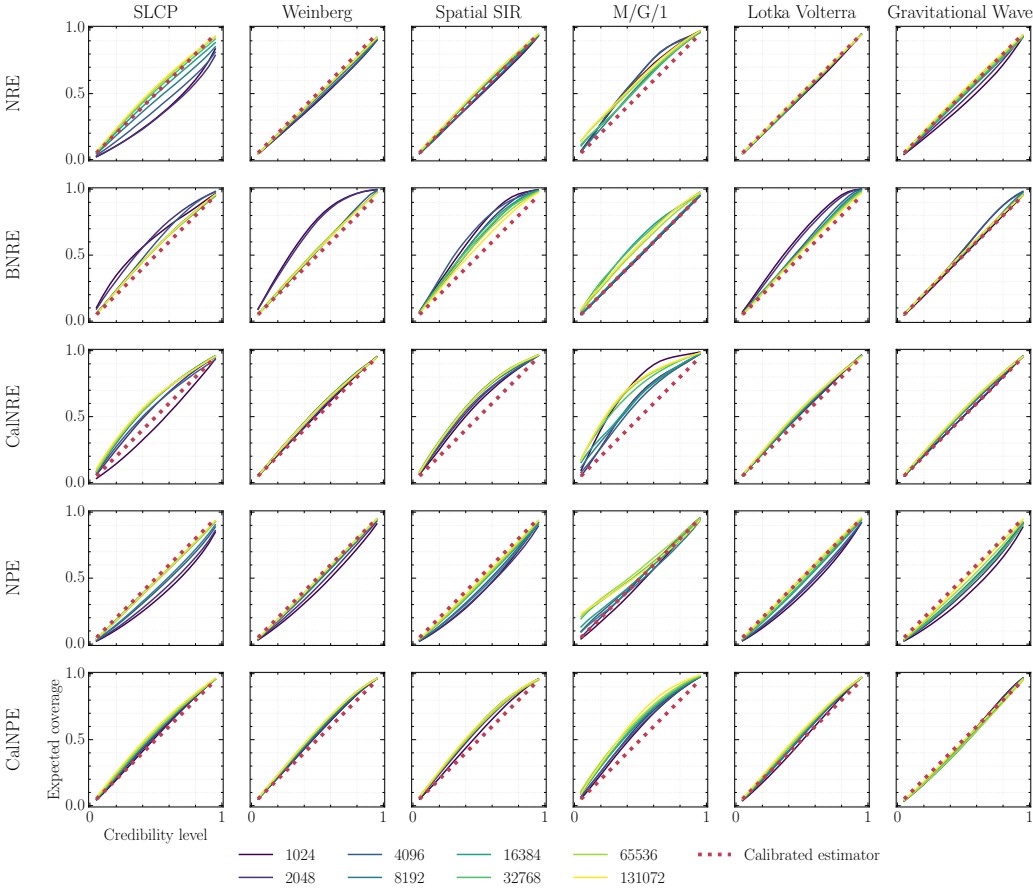

Figure 1: Evaluation on six benchmark problems (for each column) in the form of coverage curves estimated on a set of 10k test instances. If the curve is on (above) the diagonal line, then $\hat{p}(\theta|x)$ is calibrated (conservative). Expected coverage of HPDRs at 19 evenly spaced levels on the $[0.05; 0.95]$ interval is estimated following eq. (2) by solving an optimization problem to find $\Theta^{\text{HPDR}}_{\hat{p}(\theta|x)}(1-\alpha)$ for every test instance. The average of retraining five times from random initialization is shown. Top there rows marked as NRE, BNRE, and CalNRE (ours) show the NRE trained without regularization, with balancing regularizer, and the proposed regularizer with conservativeness objective, respectively. The bottom rows marked as NPE and CalNPE show the NPE trained without regularization and with the proposed regularizer with conservativeness objective, respectively. Best viewed in color.

appendix C where we report positive outcomes only for part of the studied SBI problems. As the proposal distribution for IS, we use the prior $p(\theta)$. In the main experiments, we set the weight of the regularizer $\lambda$ to 5, and the number of samples $L$ to 16 for all benchmarks. In section 4.3 we provide insights into the sensitivity of the method with respect to its hyper-parameters.

## 4.1 Principal experiments

In fig. 1, we present a cross-sectional analysis of the results of the proposed method, based on six benchmark SBI problems described in detail in Hermans et al. [20], for each considering eight different simulation budgets. Expected coverage of HPDRs at 19 evenly spaced levels on the $[0.05; 0.95]$ interval is estimated following eq. (2) by solving an optimization problem to find $\Theta^{\text{HPDR}}_{\hat{p}(\theta|x)}(1-\alpha)$ for every test instance. We intentionally use a different ECP estimation method than in the regularizer to avoid accidental overfitting to the evaluation metric. The top three rows show empirical expected coverage curves on the test set obtained by training the same model with three different objectives – NRE: classification loss; BNRE: classification loss with balancing regularizer [11]; CalNRE: classification loss with the proposed regularizer. The results for CalNRE show that the proposed method

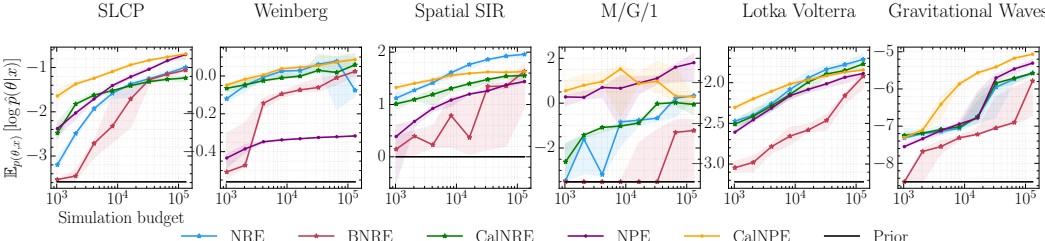

Figure 2: Expected value of the approximate log posterior density of the nominal parameters $\mathbb{E}_{p(\theta,x)}\left[\log \hat{p}(\theta|x)\right]$ of the SBI approaches in fig. 1 estimated over 10k test instances. The solid line is the median over five random initializations, and the boundaries of the shaded area are defined by the minimum and maximum values. The horizontal black line indicates the performance of the prior distribution. Best viewed in color.

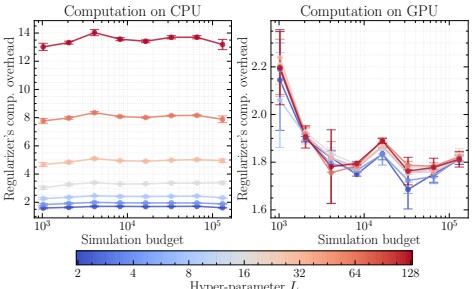

Figure 3: The computational overhead of the proposed method applied to NRE on the SLCP problem. The Y-axis indicates how many times training with the regularizer lengthens relative to training without it. For every number of samples experiments were conducted on both CPU (left chart) and GPU (right chart). Averaged over five runs, error bars show a double standard deviation range. Best viewed in color.

systematically results in conservative posteriors – with SLCP simulation budget 1024 standing out, due to too weak regularization weight $\lambda$. Moreover, in fig. 2 it can be seen that CalNRE typically outperforms BNRE in terms of $\mathbb{E}_{p(\theta,x)}\left[\log \hat{p}(\theta|x)\right]$, being close to the performance of NRE (which is not conservative), and demonstrating less variance across initializations.

The bottom two rows in fig. 1 show the same type of results for NPE-based models, with CalNPE being trained with the proposed regularizer. For all budgets in the Gravitational Waves benchmark, CalNPE does not meet the expected target, but this can easily be fixed by adjusting the regularization strength $\lambda$. In fig. 2 one finds that CalNPE sometimes yields a higher $\mathbb{E}_{p(\theta,x)}\left[\log \hat{p}(\theta|x)\right]$ than the non-regularized version, which is a counterintuitive result. We hypothesize that by regularizing the model we mitigate the classical problem of over-fitting on the training dataset.

### 4.2 Computation time

In fig. 3 we show how applying the proposed regularizer affects the training time in the case of NRE for the SLCP benchmark, with the distinction to CPU and GPU. As a reference, we take the training time of the non-regularized model. While on CPU, the training time grows about linearly with $L$, on GPU due to its parallel computation property the overhead is around x2, with some variation depending on the simulation budget and the particular run. However, this is only true as long as the GPU memory is not exceeded, while the space complexity is of order $\mathcal{O}(N\dim(\theta)L)$, where $N$ is the batch size. Comparing with fig. 2, we conclude that applying the regularizer can result in a similar $\mathbb{E}_{p(\theta,x)}\left[\log \hat{p}(\theta|x)\right]$ as in the case of increasing the simulation budget (what extends training time) while keeping comparable computational time, and avoiding over-confidence. Moreover, in situations where the acquisition of additional training instances is very expensive or even impossible, the proposed method allows for more effective use of the available data at a cost sub-linear in $L$.

### 4.3 Sensitivity analysis

In fig. 4 we show the sensitivity of the method with respect to its hyper-parameters on the example of NRE applied for the Weinberg benchmark. In fig. 4a, the analysis of regularization strength indicates a tradeoff between the level of conservativeness and $\mathbb{E}_{p(\theta,x)}\left[\log \hat{p}(\theta|x)\right]$. Therefore, choosing the

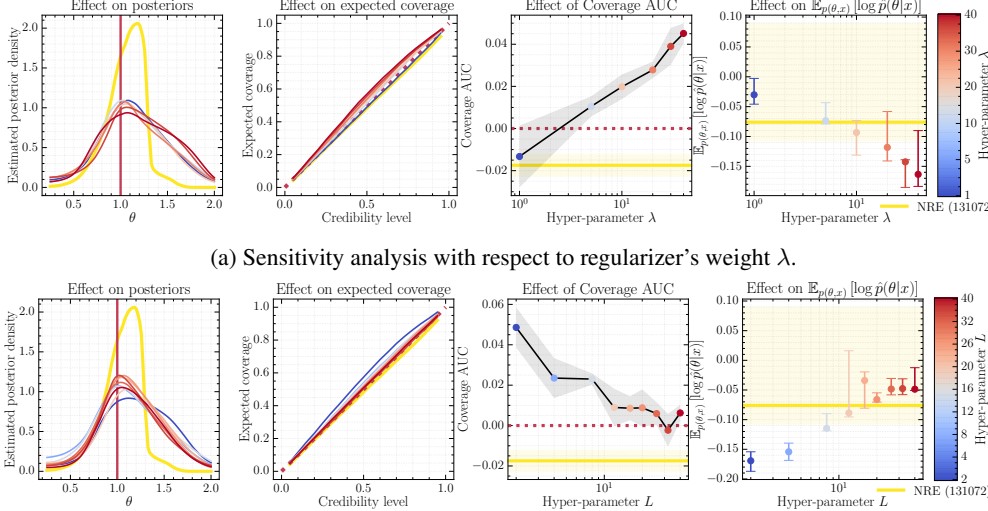

(a) Sensitivity analysis with respect to regularizer's weight $\lambda$.

(b) Sensitivity analysis with respect to the number of samples $L$ in the regularizer.

Figure 4: Sensitivity analysis study of CalNRE on the Weinberg benchmark with the smallest simulation budget of 1024 training instances with respect to $\lambda$ in (a) and $L$ in (b). The yellow color in each chart shows the results of non-regularized NRE on the highest simulation budget of 131072 training instances. Results in the first two charts show the performance of an ensemble model built from five randomly initialized members. The vertical line in the first chart marks the nominal parameter value. Points in the third chart indicate the average AUC over five randomly initialized models, with two times the standard deviation marked with the shaded area. The shaded area and the error bars in the fourth chart span between minimum and maximum results over the five randomly initialized models with the median indicated by point (line). Best viewed in color.

right configuration for practical applications is a multi-objective problem. The lowest possible $\lambda$ value that yields a positive AUC ($\lambda = 5$ in the figure) should be used to maximize predictive power as measured by $\mathbb{E}_{p(\theta,x)}\left[\log \hat{p}(\theta|x)\right]$. From fig. 4b we can read that increasing the number of samples in the regularizer brings the coverage curve closer to the diagonal while increasing $\mathbb{E}_{p(\theta,x)}\left[\log \hat{p}(\theta|x)\right]$. This suggests that a more accurate estimation of conservativeness errors during training allows better focus on the main optimization objective.

## 5 Conclusions and future work

In this work, we introduced a regularizer designed to avoid overconfident posterior distributions, which is an undesirable property in SBI. It is applicable in both the NRE and NPE settings, whenever the underlying model allows for amortized conditional density evaluation. It is derived directly from the evaluation metric based on Expected Coverage Probability, used to assess whether the posterior distribution is calibrated or conservative, depending on the target. We formally show that minimizing the regularizer to 0 error under infinite sampling guarantees the inference model to be calibrated (conservative).

We empirically show that applying the regularizer in parallel to the main optimization objective systematically helps to avoid overconfident inference for the studied benchmark problems. We admit that calculating the additional term adds computation cost at training time, but this can be mitigated with the use of modern hardware accelerators well suited for parallel computation.

The presented method is introduced in the context of simulation-based inference, but it is suitable for application in general conditional generative modeling, where calibration (conservativeness) can for example help mitigate mode collapse [50].

As a future direction, we identify the need to evaluate the proposed method in problems where the parameters' space has a high dimension. Moreover, currently, the method is restricted to neural models acting as density estimators, but not general generative models that only allow for sampling from the

learned distribution. A sampling-based relaxation of the regularizer would extend its applicability to a variety of models including Generative Adversarial Networks [33], score/diffusion-based models [52], and Variational Autoencoders [24].

## Acknowledgments and Disclosure of Funding

We acknowledge the financial support of the Swiss National Science Foundation within the MIGRATE project (grant no. 209434). Antoine Wehenkel and Arnaud Delaunoy are recipients of an F.R.S.-FNRS fellowship and acknowledge the financial support of the FNRS (Belgium). Naoya Takeishi was supported by JSPS KAKENHI Grant Number JP20K19869. The computations were performed at the University of Geneva on "Baobab" and "Yggdrasil" HPC clusters.

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

## A Proofs

**Lemma 1.** *A pair $(\theta^*, x^*)$ and a distribution $\hat{p}(\theta|x)$ uniquely define an HPDR:*

$$\Theta^{HPDR}_{\hat{p}(\theta|x)}(1 - \alpha_{HPDR}(\hat{p}, \theta^*, x^*)) := \{\theta \mid \hat{p}(\theta|x^*) \geqslant \hat{p}(\theta^*|x^*)\}, \tag{3}$$

*where*

$$\alpha_{HPDR}(\hat{p}, \theta^*, x^*) := \int \hat{p}(\theta|x) \mathbb{1}[\hat{p}(\theta|x^*) < \hat{p}(\theta^*|x^*)]d\theta. \tag{4}$$

*Proof of Lemma 1.*

$$1 - \alpha_{\text{HPDR}}(\hat{p}, \theta^*, x^*) = \int_{\Theta^{\text{HPDR}}_{\hat{p}(\theta|x)}(1 - \alpha_{\text{HPDR}}(\hat{p}, \theta^*, x^*))} \hat{p}(\theta|x)d\theta =$$

$$\int_{\{\theta|\hat{p}(\theta|x^*) \geqslant \hat{p}(\theta^*|x^*)\}} \hat{p}(\theta|x)d\theta =$$

$$\int \hat{p}(\theta|x) \mathbb{1}[\hat{p}(\theta|x^*) \geqslant \hat{p}(\theta^*|x^*)]d\theta, \quad (13)$$

while

$$\int \hat{p}(\theta|x) \mathbb{1}[\hat{p}(\theta|x^*) \geqslant \hat{p}(\theta^*|x^*)] + \int \hat{p}(\theta|x) \mathbb{1}[\hat{p}(\theta|x^*) < \hat{p}(\theta^*|x^*)] = 1 \tag{14}$$

because $\hat{p}(\theta|x)$ is a probability density function. $\qquad \square$

**Remark 1.** *An inference engine yields calibrated posterior distributions if and only if it is characterized by uniformly distributed $\alpha_{HPDR}(\hat{p}, \theta, x)s$.*

*Proof of Remark 1.* We start by observing that RHS of eq. (5) can be reformulated as $\mathbb{E}_{p(\theta, x)}[\mathbb{1}[1 - \alpha_{\text{HPDR}}(\hat{p}, \theta, x) \leqslant 1 - \alpha]]$ which is estimated using MC in the following way:

$$\frac{1}{N} \sum_{i=1}^{N} \mathbb{1}[1 - \alpha_{\text{HPDR}}(\hat{p}, \theta_i, x_i) \leqslant 1 - \alpha], \tag{15}$$

and for a calibrated model it is equal to $1 - \alpha$. The empirical ECP expressed above, takes the form of the ECDF of $1 - \alpha_{\text{HPDR}}(\hat{p}, \theta_i, x_i)$ at $1 - \alpha$. Thus, following Glivenko–Cantelli theorem we conclude that for a calibrated model, $1 - \alpha_{\text{HPDR}}(\hat{p}, \theta_i, x_i)$ has a standard uniform distribution because it is bounded in $[0, 1]$ interval by construction. Thus, the same holds for $\alpha_{\text{HPDR}}(\hat{p}, \theta_i, x_i)$.

The opposite direction is trivial to show. $\qquad \square$

### A.1 Stochastic dominance of rank statistics over $\mathcal{U}[0, 1]$ as over-confidence aversion

Let us consider the following minimization objective

$$D_N = \sup_{\alpha} |\max(F_N(\alpha) - F(\alpha), 0)|, \tag{16}$$

which leads to first-order stochastic dominance of $F_N(\alpha)$ (ECDF of $\{\alpha_{\text{HPDR}}(\hat{p}, \theta_i, x_i)\}_i^N$) over $F(\alpha)$ (target standard uniform distribution). At the minimum $D_N = 0$ we know that

$$\forall_\alpha \, \alpha \geqslant \sum_{j=1}^{N} \mathbb{1}[\alpha_{\text{HPDR}}(\hat{p}, \theta_j, x_j) \leqslant \alpha]/N. \tag{17}$$

But the LHS can be reformulated as ECDF of infinitely many samples generated with the ground-truth distribution. Without the loss of generality, assume the number of samples is equal to $N \to \infty$ on both sides.

$$\forall_\alpha \sum_{j=1}^{N} \mathbb{1}[\alpha_{\text{HPDR}}(p, \theta_j, x_j) \leqslant \alpha] \geqslant \sum_{j=1}^{N} \mathbb{1}[\alpha_{\text{HPDR}}(\hat{p}, \theta_j, x_j) \leqslant \alpha]. \tag{18}$$

With equality between LHS and RHS, this means calibrated $\hat{p}(\theta|x)$ which is a special case of conservativeness in our formulation. Let us focus only on the inequality. Then, the condition above translates into

$$\forall_\alpha \exists_j \; \alpha_{\text{HPDR}}(p, \theta_j, x_j) \leqslant \alpha \; \& \; \alpha_{\text{HPDR}}(\hat{p}, \theta_j, x_j) > \alpha, \tag{19}$$

thus

$$\forall_\alpha \exists_j \; \alpha_{\text{HPDR}}(p, \theta_j, x_j) < \alpha_{\text{HPDR}}(\hat{p}, \theta_j, x_j). \tag{20}$$

Now, insert the definition of $\alpha_{\text{HPDR}}(\cdot, \theta_j, x_j)$:

$$\forall_\alpha \exists_j \sum_{k=1}^{L} \mathbb{1}\left[\hat{p}(\theta^*_{(j)k}|x_j) < \hat{p}(\theta_j|x_j)\right] < \sum_{k=1}^{L} \mathbb{1}\left[\hat{p}(\theta_{(j)k}|x_j) < \hat{p}(\theta_j|x_j)\right], \tag{21}$$

where $\theta^*_{(j)k}$ are sampled from the ground truth posterior $p(\theta|x)$, and $\theta_{(j)k}$ from the approximate posterior $\hat{p}(\theta|x)$. The following two steps show equivalent conditions:

$$\forall_\alpha \exists_j \exists_k \; \hat{p}(\theta^*_{(j)k}|x_j) \geqslant \hat{p}(\theta_j|x_j) \; \& \; \hat{p}(\theta_{(j)k}|x_j) < \hat{p}(\theta_j|x_j), \tag{22}$$

$$\forall_\alpha \exists_j \exists_k \; \hat{p}(\theta^*_{(j)k}|x_j) > \hat{p}(\theta_{(j)k}|x_j). \tag{23}$$

The condition in eq. (23) says that ground-truth posterior generates at least one sample with higher density measured by the approximate posterior $\hat{p}(\theta|x)$ than that generated by the approximate posterior. Thus, we have shown that a model generating $\alpha_{\text{HPDR}}(\hat{p}, \theta_i, x_i)$s stochastically dominating standard uniform distribution has an aversion to over-confidence.

## A.2 Convergence to uniform

Let us[2] consider a discrete uniform distribution over $L+1$ evenly spaced values on the $[0, 1]$ interval, which has the following CDF:

$$F_L(x) = \begin{cases} 0, & x < 0 \\ \lfloor xL \rfloor / L, & 0 \leqslant x < 1 \\ 1, & x \geqslant 1 \end{cases}$$

The CDF of the standard continuous uniform distribution has the following form:

$$F(x) = \begin{cases} 0, & x < 0 \\ x, & 0 \leqslant x < 1 \\ 1, & x \geqslant 1 \end{cases}$$

For $0 \leqslant x < 1$ we have

$$|F(x) - F_L(x)| = |x - \lfloor xL \rfloor / L| = |xL - \lfloor xL \rfloor| / L < 1/L,$$

where the inequality is a property of the floor $\lfloor \cdot \rfloor$ operator.

Taking the limit of $L \to \infty$ we have

$$\lim_{L \to \infty} |F(x) - F_L(x)| = \lim_{L \to \infty} 1/L = 0,$$

what proofs convergence in the distribution of $F_L(x)$ to $F(x)$.

## A.3 GC theorem inverse direction

We[3] are given a set of iid samples $\alpha_1, \ldots, \alpha_N$ from a $G$ distribution implicitly given by the trained model. With the regularizer optimized to $0$ loss, we assume that

$$D_N = \sup_\alpha |F_N(\alpha) - F(\alpha)| \overset{a.s.}{\to} 0.$$

---

[2]We follow the derivation from `https://stats.stackexchange.com/a/373350` (version: 2018-10-23).
[3]We follow the derivation from `https://stats.stackexchange.com/q/485148` (version: 2020-08-29).

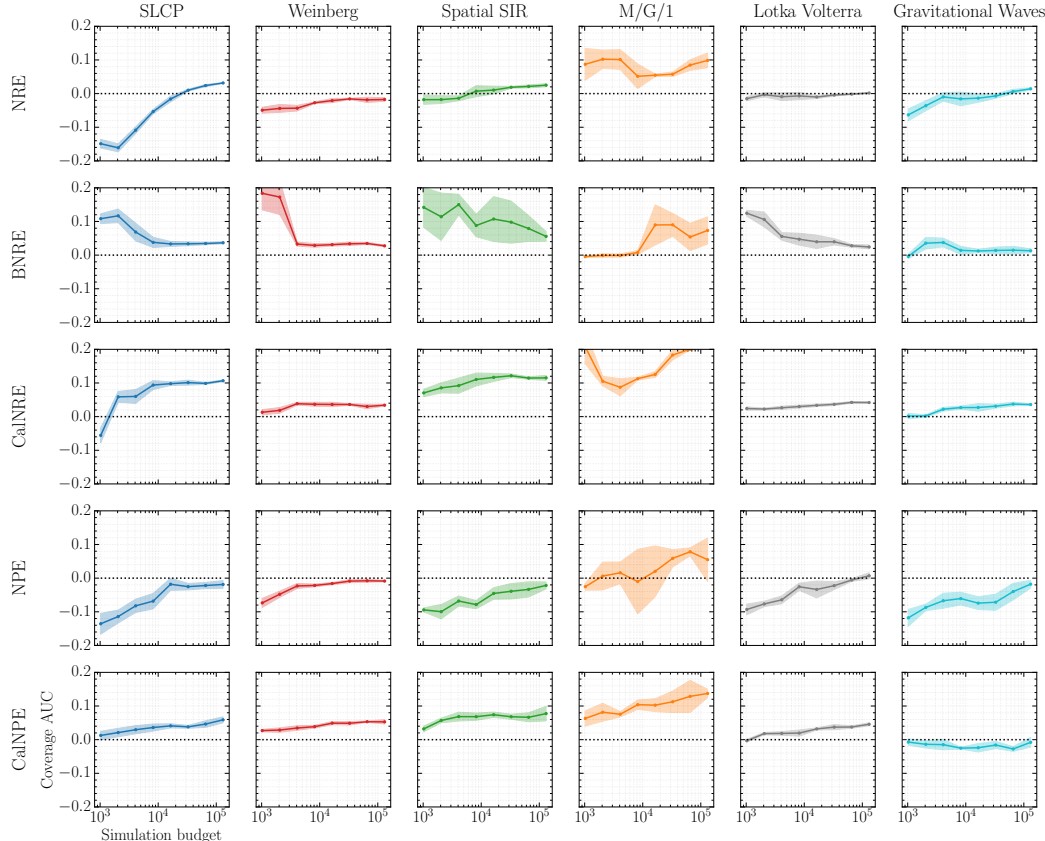

Figure 5: AUC scores of the coverage curves presented in fig. 1. Points indicate the average, while the shaded area spans over the two standard deviation range. Best viewed in color.

From the Glivenko-Cantelli theorem, we have that

$$D_N = \sup_\alpha |F_N(\alpha) - G(\alpha)| \overset{a.s.}{\to} 0.$$

The triangle inequality is satisfied for any norm, so:

$$\sup_\alpha |F_N(\alpha) - F(\alpha)| + \sup_\alpha |F_N(\alpha) - G(\alpha)| > \sup_\alpha |G(\alpha) - F(\alpha)|,$$

thus

$$\sup_\alpha |G(\alpha) - F(\alpha)| \overset{a.s.}{\to} 0,$$

where the LHS does not depend on $N$, and we conclude that $G = F$, what was to be shown.

## B    Additional conservative posterior results

In this section, we supplement the results shown in section 4.1 by presenting the coverage AUC scores in fig. 5. We show explicitly, that applying the proposed method with a conservativeness objective leads to conservative posteriors. For the two non-conservative cases – CalNRE for SLCP, and CalNPE for Gravitational Waves – we expect a positive result when choosing the appropriate value of the regularization weight, as already stated in the main text.

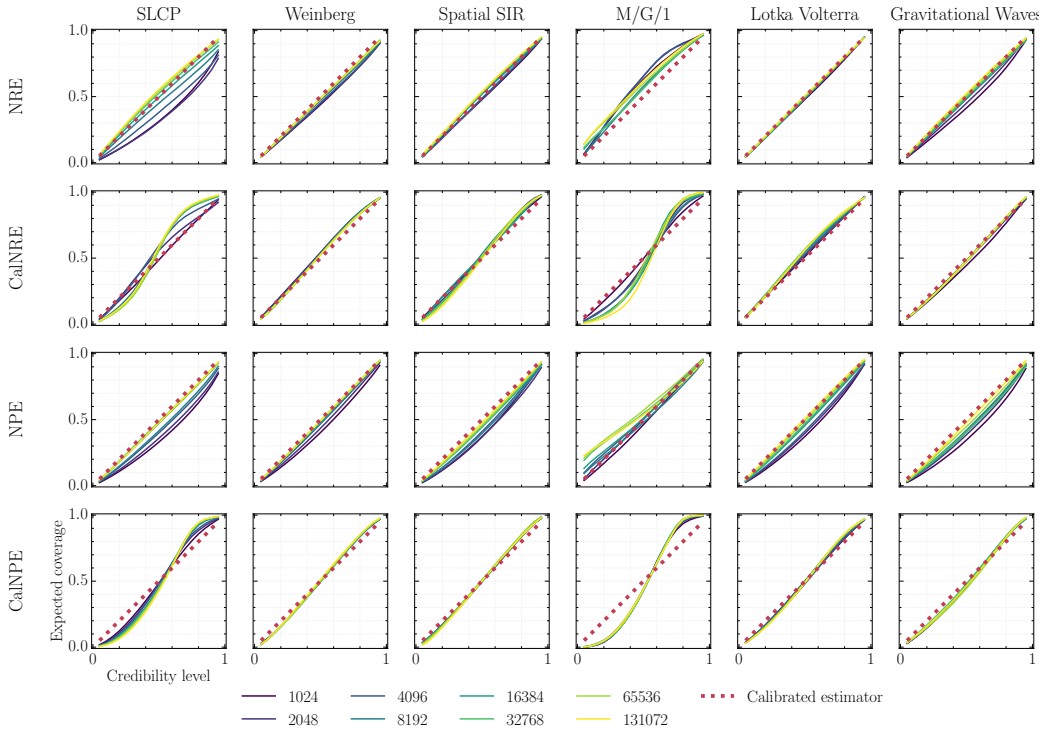

Figure 6: Evaluation on six benchmark problems (for each column) in the form of coverage curves estimated on a set of 10k test instances. If the curve is on (above) the diagonal line, then $\hat{p}(\theta|x)$ is calibrated (conservative). Expected coverage of HPDRs at 19 evenly spaced levels on the $[0.05; 0.95]$ interval is estimated following eq. (2) by solving an optimization problem to find $\Theta^{\text{HPDR}}_{\hat{p}(\theta|x)}(1 - \alpha)$ for every test instance. The average of retraining five times from random initialization is shown. Top two rows marked as NRE, and CalNRE (ours) show the NRE model trained without regularization, and the proposed regularizer with calibration objective, respectively. The bottom rows marked as NPE and CalNPE show the NPE model trained without regularization and with the proposed regularizer with calibration objective, respectively. Best viewed in color.

## C  Regularization for calibrated posterior

In this section, we present the results of experiments analogous to those in section 4.1, but with the proposed regularizer configured as calibration error (see eq. (7)). Obtaining a calibrated posterior is a task way more difficult than in the case of a conservative one, and this is what we empirically confirm.

In fig. 6 we find a characteristic S-shaped coverage curve emerging for the regularized models. For the low credibility levels (small credible regions) posteriors become overconfident, while with the high levels (big credible regions) they become conservative, by a certain margin. Overall, this results in an approximately 0 AUC as shown in fig. 8. However, it is only by chance that it becomes positive (on average conservative), and not negative (on average overconfident). From fig. 7 we also see that the predictive power of the regularized models measured by $\mathbb{E}_{p(\theta,x)}\left[\log \hat{p}(\theta|x)\right]$ estimated on the test set degrades compared to the conservativeness setting in fig. 2.

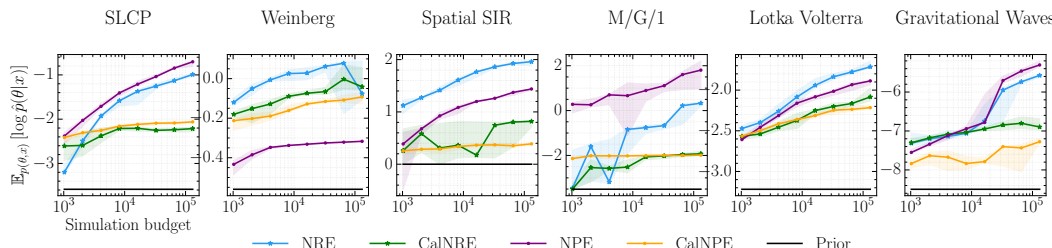

Figure 7: Expected value of the approximate log posterior density of the nominal parameters $\mathbb{E}_{p(\theta,x)}\left[\log \hat{p}(\theta|x)\right]$ of the models in fig. 6 estimated over 10k test instances. The solid line is the median over five random initializations, and the boundaries of the shaded area are defined by the minimum and maximum values. The horizontal black line indicates the performance of the prior distribution. Best viewed in color.

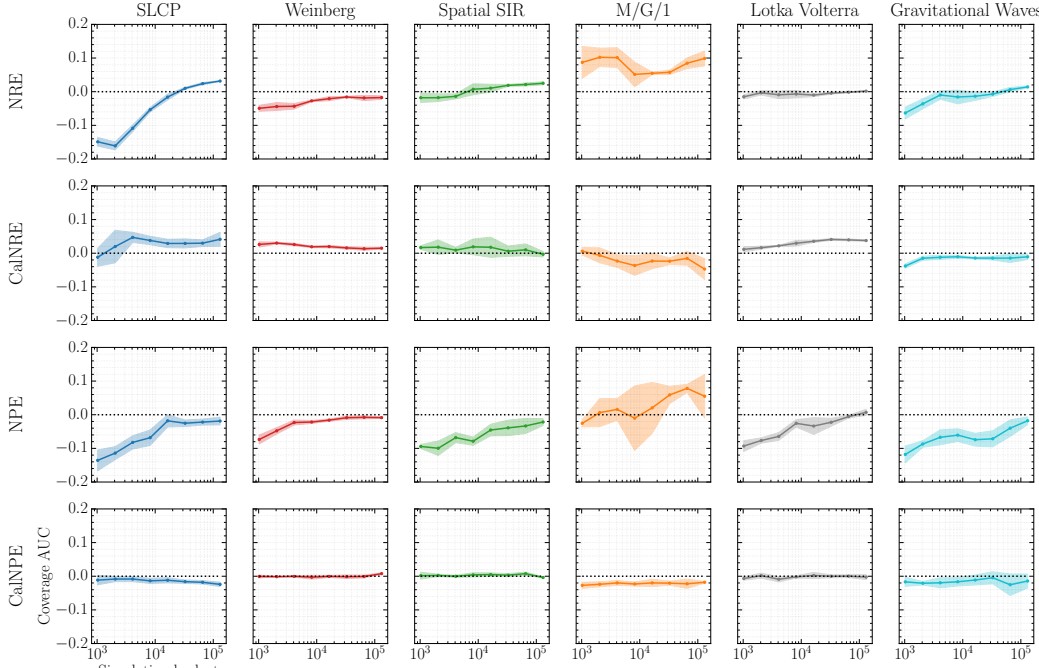

Figure 8: AUC scores of the coverage curves presented in fig. 6. Points indicate the average, while the shaded area spans over the two standard deviation range. Best viewed in color.

## D    Intuition about coverage analysis

In this section, we build a toy example to intuitively introduce the concept of Expected Coverage Probability and the over-/under-confidence that results from it. We rely on non-conditional one-dimensional distributions for clarity. In fig. 9 we introduce a black multimodal ground-truth distribution and a red one intentionally under-dispersed, but with the same modes, acting as the approximate distribution. We find the $1 - \alpha = 0.9$ Highest Density Region for the red distribution, which consists of two segments concentrated around the modes. To mimic the real-world scenario we sample $N = 1000$ points from the ground-truth distribution and use them to estimate the expected coverage probability, which is trivial in one dimension knowing the segments' boundaries. According to the plan, the coverage analysis showed that the red distribution is overconfident for evidence from the data.

## E    Implementation details

For the details of the architecture of Neural Ratio Estimation models please refer to Delaunoy et al. [11]. The SELU function [25] is used as an activation function in the feed-forward networks.

For the details of the architecture of Neural Posterior Estimation models please refer to Hermans et al. [20].

All the models with the proposed regularizer are trained for $500$ epochs with $128$ batch size with AdamW optimizer with a $0.001$ learning rate.

Additional remarks:

- We use the following PyTorch implementation of Blondel et al. [6]: `https://github.com/teddykoker/torchsort`. We use it with the default parameters.
- In order to stabilize the training of regularized NRE (CalNRE) we introduced Gradient Norm Clipping.
- Training of CalNPE for the M/G/1 in the conservativeness setting with the last four simulation budgets was challenging due to extreme values in observations. It is reflected in the decrease of $\mathbb{E}_{p(\theta,x)}\left[\log \hat{p}(\theta|x)\right]$ in fig. 2 as the simulation budget increases.
- When computing the regularizer the same $\hat{p}(\cdot|x^*)$ is repeatedly evaluated, therefore, our implementation evaluates $x^*$s embedding only once (per regularizer evaluation) and reuses it for all samples.

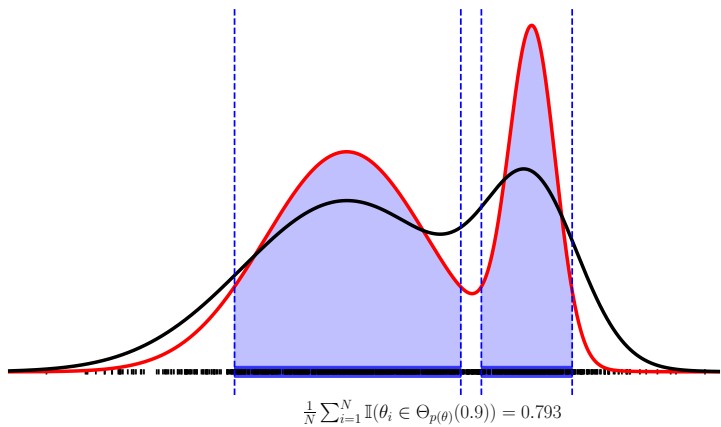

$$\frac{1}{N}\sum_{i=1}^{N}\mathbb{I}(\theta_i \in \Theta_{p(\theta)}(0.9)) = 0.793$$

Figure 9: Plots of two probability density functions of 2-mixture of Gaussians (0.7/0.3 ratio), with the red one ($\sigma_1 = 0.7$, $\sigma_2 = 0.2$) being under-dispersed compared to the black one ($\sigma_1 = 0.9$, $\sigma_2 = 0.4$). The blue region marks 0.9 Highest Posterior Density Region of the red distribution. The black points on the x-axis are random samples from the black distribution. Below the graph, the empirical Expected Coverage Probability of the red distribution against $N = 1000$ samples of the black distribution is computed. Best viewed in color.

