# OpenReview forum: "Calibrating Neural Simulation-Based Inference with Differentiable Coverage Probability"
_NeurIPS.cc/2023/Conference — NeurIPS 2023 poster_

### Official Review · Reviewer_JSnb · 2023-07-02

**Soundness:** 4 excellent
**Presentation:** 2 fair
**Contribution:** 3 good
**Rating:** 6
**Confidence:** 4

**Summary:**

The paper presents a method to perform calibrated simulation-based inference. To do so, the paper employs the well-known coverage and proposes a way to differentiate through this term and to use it as a regularizer during training. The authors evaluate their method on benchmark tasks and conclude that it has good coverage and (sometimes) even outperforms existing methods in terms of log-likelihood.

**Strengths:**

**Originality**:
The method is novel and the use of a differentiable sorting algorithm for differentiating through coverage is novel and useful.

**Quality**:
The theoretical part of the paper is done rigorously and the paper provides additional empirical results for the impact of hyperparameters and computational cost.

**Clarity**:
The figures are clear and support the messages of the paper.

**Weaknesses:**

**Quality**:
I expect that the method is very expensive if the batch size is large because in this case, GPU will not help either. This should be clarified in the paper.

I found it very interesting that CA1NPE outperforms NPE in terms of log-likelihood. What exactly are the methods that the authors use to prevent overfitting? Always training for 500 epochs is clearly not something anybody would do in practice (L497). Please use a proper implementation of NRE or NPE to draw comparisons.

The poor results of Appendix Fig 6 should be mentioned in the main paper and it should be highlighted that the method can be used to produce conservative posteriors, but that it is not suitable to produce calibrated posteriors.

All tasks in the paper are very low dimensional. I would expect the NPE version of this algorithm to scale to high-dimensional asks, but this would need emprical evidence. For NRE, I could imagine that Importance sampling requires exceedingly many samples in high-d parameter spaces. Please clarify and ideally add tasks with a more high-dimensional parameter space.


**Clarity**:

Twice (L159 and L183), the authors propose alternative formulations of their method. They never empirically investigate these formulations and also do not describe why they are less good. I would appreciate if the authors either add additional details on these methods or remove them entirely to avoid confusion.

The paper introduces **many** symbols which makes the paper very tedious to read. I would appreciate if the authors would redefine symbols in new sections to make the paper easier to follow. Also, some abbreviations do not really have to be defined (rarely used, e.g. KS, SAE, STE).

Many papers are listed as `arxiv` although they got published. Please fix.


**Questions:**

Why is the sorting based-computation preferrable over the direct computation? The authors say that `we need to backpropagate through F_N(\alpha_k)`. Is this a problem in practice?

L472: what do you mean by “proposed regularizer set to calibration objective”

L505: what does “outstanding values in observations” mean?

**Limitations:**

The authors state limitations of their method, but some limitations have to be highlighted (see comments above, compute time for large batch sizes, poor performance for calibrated (not convservative) posteriors.

---

> ### Author Rebuttal · Authors · 2023-08-08
>
> Dear Reviewer,
>
> Thank you very much for taking the time to review our manuscript and for your comments. Below we would like to address the questions:
>
> - *I expect that the method is very expensive if the batch size is large because in this case, GPU will not help either. This should be clarified in the paper.*
>
>    In L262 we say that the advantage of computation on GPU is due to the parallelization. Implicitly, this means that once <parameters dimensionality X batch size X number of samples $L$> exceeds the available memory, parallelization, and consequently GPU advantage is limited. We will explicitly mention this threat in the revised version of the manuscript.
>
>
> - *I found it very interesting that CA1NPE outperforms NPE in terms of log-likelihood. What exactly are the methods that the authors use to prevent overfitting? Always training for 500 epochs is clearly not something anybody would do in practice (L497). Please use a proper implementation of NRE or NPE to draw comparisons.*
>
>    We use the same training protocol for both NPE and CalNPE that was used in Hermans et al., 2022 - literally, the same implementation that is available online. Although we run 500 epochs of training in total, only the best model based on validation log-likelihood is kept as the final model. In addition, Gradient Norm Clipping is used. We consider the implementation from Hermans et al., 2022 to be proper (which is based on some standard packages used in the field of SBI). Moreover, both the regularized and non-regularized versions use the same implementation allowing us to draw conclusions.
>
>
> - *The poor results of Appendix Fig 6 should be mentioned in the main paper and it should be highlighted that the method can be used to produce conservative posteriors, but that it is not suitable to produce calibrated posteriors.*
>
>    We will add this information in the revised version of the manuscript.
>
>
> - *All tasks in the paper are very low dimensional. I would expect the NPE version of this algorithm to scale to high-dimensional asks, but this would need empirical evidence. For NRE, I could imagine that Importance sampling requires exceedingly many samples in high-d parameter spaces. Please clarify and ideally add tasks with a more high-dimensional parameter space.*
>
>    Following the nowadays state-of-the-art evaluation proposed in Hermans et al., 2022 already imposes a huge computational effort. We see the high-d parameter spaces as an important challenge and list it as a future work direction. However, with the available computational resources we are unable to provide empirical evidence in the short term.
>
> - *Many papers are listed as arxiv although they got published. Please fix.*
>
>    In the revised version of the manuscript, we will update all the references that got published since we submitted the manuscript.
>
> - *L472: what do you mean by “proposed regularizer set to calibration objective”*
>
>    Minimizing calibration error not the conservativeness error. We will clarify this information in the revised version of the manuscript.
>
>  - *L505: what does “outstanding values in observations” mean?*
>
>    The experimental protocol of Hermans et al., 2022 (which we follow) did not include data standardization. The highest value encountered in observations for the largest simulation budget of M/G/1 was 4e+7 which is the “outstanding value” in L505. However, it was unlikely to happen, and therefore for the small simulation budgets, the extreme observations were less extreme leading to a stable learning process. We will replace “outstanding value” with “extreme value” to avoid confusion.

---

> > ### Comment · Reviewer_JSnb · 2023-08-13
> > **Response to rebuttal**
> >
> > Thank you very much for the detailed response. It clarifies most of my concerns, and I have increased my score to 6.
> >
> > In my opinion, the main limitation remains that the method is only being demonstrated on very low-dimensional parameter spaces, in particular because importance sampling could lead to issues in more high-D parameter spaces. It would be amazing if the authors could add such analyses for the camera ready version, but I do think that this is a strong paper either way. Congrats on this nice work!

---

### Official Review · Reviewer_KAKy · 2023-07-03

**Soundness:** 3 good
**Presentation:** 4 excellent
**Contribution:** 3 good
**Rating:** 7
**Confidence:** 3

**Summary:**

The paper proposes a calibration term to be used directly in the training objective of NREs and NPEs. The paper shows that the introduction of this term achieves competitive or better results in terms of coverage and expected posterior density.

**Strengths:**

* The quality of the writing and presentation is high, with the structure easy to follow. In particular the related work is clearly included in the introduction.
* The topic is of importance as it is increasingly more common to use neural estimators in SBI applications.
* The experimentation is sufficient to show the effect of incorporating the new regulariser. It seems to show that the new regulariser leads to slightly conservative estimators that are on average better calibrated than the baselines.


**Weaknesses:**

* The paper does not appear to have any major weakness. A few minor weaknesses that seem to exist have been appropriately described in the paper. These are the computational cost of the approach and the fact that the paper’s main metric of performance is also the same one that has been used in the regulariser. If the authors could think of a different metric to use to evaluate performance then the paper would be further strengthened. However, it is appreciated that finding a new metric would potentially be a new paper in its own right. Perhaps including C2ST as a metric in the appendix might provide additional comparison. While the focus of the paper is on calibration, it is possible to get a perfectly calibrated, but badly performing estimator.
* There is no mention of hyperparameter optimisation (although a sensitivity analysis is given).


**Questions:**

* Line 206: Does IS provide an improvement for NPE, even when it is not needed?
* Since the regulariser relies on importance sampling did the authors do any experimentation on how the approach scales with dimension of $\theta$? This was mentioned in future work, but even in the experiments in the paper the theta varies across experiments. What are the dimensions of the experiments included in this paper? And did the authors see that the higher dimensional ones performed worse with the new regulariser?


**Limitations:**

* The limitation related to computational cost is sufficiently highlighted in the work.

---

> ### Author Rebuttal · Authors · 2023-08-08
>
> Dear Reviewer,
>
> Thank you very much for taking the time to review our manuscript and for your comments. Below we would like to address the questions:
>
> - *Line 206: Does IS provide an improvement for NPE, even when it is not needed?*
>
>    Intuition suggests that IS should introduce noise and worsen the results. In a limited study (please see Figure 3 of the global response PDF) we did a sanity check on two problems (Weinberg - 1D posterior; Lotka Volterra - 2D posterior) where IS is not used for NPE. The results show that IS with 16 samples (submitted manuscript) gives almost the same outcomes as directly sampling from the approximate posterior with the reparameterization trick (Figure 3 in global response PDF). We expect the comparison to look very different for moderate and high-dimensional posteriors.
>
>
> - *Since the regulariser relies on importance sampling did the authors do any experimentation on how the approach scales with dimension of ? This was mentioned in future work, but even in the experiments in the paper the theta varies across experiments. What are the dimensions of the experiments included in this paper? And did the authors see that the higher dimensional ones performed worse with the new regulariser?*
>
>    We did experiments only with the problems mentioned in the submitted manuscript. The dimensions of posteriors in the submitted manuscript are as follows: SLCP - 2D; M/G/1 - 3D; Weinberg - 1D; Lotka Volterra - 2D; Spatial SIR - 2D; Gravitational Waves - 2D. All of the studied problems are of the same order of magnitude in terms of the dimensionality of parameters (which is not the case for the dimensionality of observations but this has limited impact on the regularizer), therefore we cannot draw conclusions about scaling. We hold that this should be verified in subsequent work.

---

> > ### Comment · Reviewer_KAKy · 2023-08-14
> > **Response**
> >
> > Thanks - confirming that I have read the rebuttal.

---

### Official Review · Reviewer_Lpwd · 2023-07-06

**Soundness:** 3 good
**Presentation:** 4 excellent
**Contribution:** 3 good
**Rating:** 7
**Confidence:** 4

**Summary:**

The authors suggest a new objective function for simulation-based inference that adds a penalty term to the “expected score” objective that is used by many other works. This penalty term encourages the resulting posterior approximations to be well-calibrated in the sense that the $1-\alpha$ highest probability density regions contain the ground-truth parameter $100(1-\alpha)$% of the time. The penalty term is an extension of the Kolmogorov-Smirnov test statistic for goodness of fit between a $U(0,1)$ distribution and the rank statistics of eq. 7, which are asymptotically uniformly distributed when $p(\theta \mid x)$ is calibrated. Minimizing this penalty term alongside the usual objective function should yield posteriors that are calibrated while substantially different from the degenerate case of the prior (which is trivially calibrated).


**Strengths:**

* The proposed penalty/regularization term is relatively lightweight computationally, and can be tacked on to many existing simulation-based inference algorithms.
* The intuition is simple; the construction of the statistics $\hat{\alpha}$ is clever, and the Kolmogorov-Smirnov test statistic is well-understood.
* The method is assessed on a variety of problems from the test suite of benchmarks provided by Lueckmann et al. (2021), and performs favorably in that the resulting posteriors are either calibrated or tend to be conservative.


**Weaknesses:**

* Although synthetic likelihood and ABC approaches are mentioned in the introduction, the proposed method seems prohibitively costly in these scenarios, and seems geared toward amortized methods only, where eq. 7 can be computed rapidly.
* For ratio estimation in particular, the non-regularized NRE is better calibrated than the corrected version in some cases (e.g., spatial SIR), suggesting that it is at least possible the penalty term can worsen calibration.
* In practice, it appears that attempts at calibration often result in conservative rather than calibrated posteriors; while this is likely still preferable to many users to the alternative of *not* adding this correction, it suggests a bit of a misnomer.


**Questions:**

* The method called neural posterior estimation (NPE) has been shown in other work to be equivalent to minimizing a forward KL divergence between exact and approximate posteriors (Reweighted Wake-Sleep (Bornschein and Bengio, 2015), and a similar work “Revisiting Reweighted Wake Sleep…” (Le et al., 2018)); it’s been argued in these that the resulting neural posterior estimates tend to be overdispersed or conservative as a result. 1) How does this impact the relevance of this work if the baseline is usually already conservative? 2) The NPE row of Figure 1 doesn’t seem to reflect this intuition in practice; is there any explanation for why this is so? Is NPE indeed the same as sleep-phase in reweighted wake-sleep in the experiments?
* What are the advantages and disadvantages of using either the sorting-based computation or the direct computation method? Is the use of the latter solely due to the belief that backprop through indicator functions is somehow worse? As $\hat{\alpha}$ itself requires indicator functions, I suppose the backprop operation can’t be too problematic. It would be interesting to see both methods implemented in the experiments.
* The form of eq. 9 suggests self-normalized importance sampling, but as $\hat{p}(\theta \mid x^*)$ is already normalized, this doesn't seem necessary. Maybe this is a misunderstanding of notation. To my reading, though, just the numerator of eq. 9 is a valid importance sampling estimator. Is so-called “standard” or self-normalized importance sampling nonetheless used?


**Limitations:**

Generally the settings where this work may be applied are clear, but as mentioned above discussing the use of the method in amortized vs. non-amortized settings may make more clear the costs associated with with this method when $\hat{p}(\theta_j \mid x^*)$ can’t be computed quickly.

---

> ### Author Rebuttal · Authors · 2023-08-08
>
> Dear Reviewer,
>
> Thank you very much for taking the time to review our manuscript and for your comments. Below we would like to address the questions:
>
> - *Although synthetic likelihood and ABC approaches are mentioned in the introduction, the proposed method seems prohibitively costly in these scenarios, and seems geared toward amortized methods only, where eq. 7 can be computed rapidly.*
>
>    We would like to clarify that the proposed method is intended only for amortized methods. Synthetic likelihood and ABC are mentioned only to give a more complete context of SBI, we will make that more clear in the text.
>
>
> - *For ratio estimation in particular, the non-regularized NRE is better calibrated than the corrected version in some cases (e.g., spatial SIR), suggesting that it is at least possible the penalty term can worsen calibration.*
>
>    The non-regularized NRE for Spatial SIR is not conservative for the first three simulation budgets, while CalNRE is conservative for all of the analyzed budgets which is the improvement that we intend to introduce with the proposed method. Indeed, the distance of the coverage curve from the diagonal may increase when using the proposed regularization term, but this is not a violation of the objective in eq. (6) re-formulated in L127 for conservativeness.
>
>
> - *In practice, it appears that attempts at calibration often result in conservative rather than calibrated posteriors; while this is likely still preferable to many users to the alternative of not adding this correction, it suggests a bit of a misnomer.*
>
>    We would like to clarify that the results presented in the main part of the submitted manuscript (Figures 1 & 2 + Figure 5 in the Appendix) are obtained with the objective of minimizing conservativeness error (L130), not calibration error. The result of the latter, we show in the Appendix (Figures 6-8) and find it difficult to draw any firm conclusions. Finding a calibrated approximate posterior is a very difficult task and we make no claim that the proposed method is able to systematically provide it. We will emphasize this in the revised version of the manuscript.
>
>
> - *The method called neural posterior estimation (NPE) has been shown in other work to be equivalent to minimizing a forward KL divergence between exact and approximate posteriors (Reweighted Wake-Sleep (Bornschein and Bengio, 2015), and a similar work “Revisiting Reweighted Wake Sleep…” (Le et al., 2018)); it’s been argued in these that the resulting neural posterior estimates tend to be overdispersed or conservative as a result. 1) How does this impact the relevance of this work if the baseline is usually already conservative? 2) The NPE row of Figure 1 doesn’t seem to reflect this intuition in practice; is there any explanation for why this is so? Is NPE indeed the same as sleep-phase in reweighted wake-sleep in the experiments?*
>
>    The training objective of NPE is indeed the same as the one used in sleep-phase of RWS, but with instances $(\theta, x)$ coming from a fixed training dataset (measured in the field or sampled from a simulator) in contrast to “dream samples” from the trained generative network in RWS. So the problem solved by these two methods is fundamentally different.
>
>    The NPE row of Figure 1 is reprinted from Hermans et al., 2022 where in an intense empirical study the authors identify multiple methods (including NPE) to be over-confident (non-conservative) for a number of benchmark problems in SBI – in a limited study we confirmed their results (not reported in the submitted manuscript). In our work, we use the same benchmark problems, and with the application of the proposed regularizer are able to train models that are conservative in cases where the non-regularized models were not. We do not claim that the use of our method is necessary, but rather we propose it as a solution when over-confidence is identified. In addition, we would like to note that Hermans identifies two standard solutions in machine learning as often effective in the analyzed problems, i.e. increasing the amount of the training data and ensembling. Therefore, it is enough that in the cited works in the field of RWS, the budget for simulations during training was large enough that the problem of over-confidence did not arise. Meanwhile, in SBI, obtaining a sufficient dataset can be very expensive, or even impossible.
>
>
> - _The form of eq. 9 suggests self-normalized importance sampling, but as $\hat{p}(\theta | x^*)$ is already normalized, this doesn't seem necessary. Maybe this is a misunderstanding of notation. To my reading, though, just the numerator of eq. 9 is a valid importance sampling estimator. Is so-called “standard” or self-normalized importance sampling nonetheless used?_
>
>    Indeed, the self-normalized importance sampling is given in eq. (9) and also used in our experiments. For NPE the denominator is completely unnecessary because the approximate posterior is normalized by construction. However, in the case of NRE, there is no such guarantee (we only hope that this is achieved at convergence) which exposes a slight abuse of notation on our side. Therefore, self-normalized IS is advisable for NRE and to have a single implementation we use it also for NPE.

---

> > ### Comment · Reviewer_Lpwd · 2023-08-16
> > **Rebuttal response**
> >
> > Thanks to the authors for the detailed point-by-point response. My main question about the instances of over-confidence (i.e. lack of conservativeness) for NPE has been answered by the use of a fixed dataset for training. As this has been reproduced from and discussed extensively in Hermans et al., I’m satisfied with these. The authors have indicated that they will make more clear the distinction between amortized and non-amortized approaches as well, which is appreciated.
> >
> > My primary remaining critique is on the use of “calibration” as the selling point of the paper. While I agree that conservative posteriors are beneficial, they can also be obtained trivially (e.g., take the prior), and I’m not sure I agree with the authors that “... we make no claim that the proposed method is able to systematically provide [calibrated posteriors]” due to the title. The authors have indicated that they will address this in the exposition and I hope this is done.
> >
> > A new concern I have noticed is that in the experiments, a higher computational budget seems to result in more conservative posteriors sometimes. In particular, I’m looking at the M/G/1 column of Figure 1 in the main body. In the “Cal” versions, the yellow line corresponding to the highest computational budget is either the most or second-most conservative on average. I would appreciate the authors’ comments on this point. This compares unfavorably with, say, BNRE which seems to maintain conservativeness, while becoming more and more calibrated as the computational budget increases (e.g., in the SIR plot for BNRE).

---

> > > ### Author Response · Authors · 2023-08-18
> > >
> > > - "...higher computational budget seems to result in more conservative posteriors sometimes..."
> > >
> > > To address the question we will use Figure 5 located in the B section of the Appendix. There, one finds that BNRE indeed tends to yield AUC closer to zero as the simulation budget increases, while our proposed regularizer (CalNRE and CalNPE rows) does not exhibit such a behavior.
> > >
> > > We consider it to be consistent with the properties of the proposed solution when the conservativeness error is minimized (results presented in Fig 1, 2, and 5), because once the model yields conservative posteriors there is no penalty from the regularizer, and thus the training objective does not distinguish between "more conservative" and "less conservative" models. It turns out, that in such a regime the main loss term favors less-confident models when exposed to more training data. This may seem counter-intuitive at first (maximizing predictive performance -> minimizing under-confidence) but is the desired phenomenon when the inverse problem is ambiguous (ground-truth posterior is not a Dirac delta), because a higher simulation budget means exposure to more diverse samples that should result in less-confident models. In fact, this is exactly what we observe for the non-regularized models, therefore we conclude this is a behavior of the NRE/NPE approach itself.
> > >
> > > If bringing the AUC as close to zero as possible is desired we hypothesize that adding a low-weighted calibration error term in the late phase of training (or adding it progressively) could bring some improvement. However, we did not conduct such experiments and now see it as a potential future work direction.

---

### Official Review · Reviewer_T8qr · 2023-07-27

**Soundness:** 3 good
**Presentation:** 3 good
**Contribution:** 3 good
**Rating:** 7
**Confidence:** 3

**Summary:**

The paper introduces a training algorithm for posterior distribution learning in the likelihood free setting that combats overconfident models.  The authors focus on the expected coverage probability (ECP) from Hermans et al., 2022 to measure if a model posterior is conservative.    Specifically, when its value is equal to the credibility level used in its calculation, then the posterior is calibrated.  An equivalent condition is that the distribution of credibility levels constructed from samples (lemma 1) is uniformly distributed.  With this in mind, the authors propose a regularizer that penalizes how far the distribution of these credibility levels are from a uniform distribution.   The main contribution of the paper is the algorithm used to train with this regularizer.  The authors use a variety of techniques in their algorithm including a one-sample Kolmogorov-Smirnov test to work with finite samples, differentiable sorting to implement the test, and importance sampling to choose useful samples to use.  The experimental results demonstrate that the learned models are more conservative than their unregularized counterparts while not sacrificing performance in terms of likelihood.  Furthermore, the authors showed the effect of the hyperparameters on performance and also showed the that the training algorithm can be expensive if not run on GPUs.

**Strengths:**

- Directly addresses the "trust crisis in simulation-based inference" described in Hermans et al., 2022.
- Strong empirical results to demonstrate that the training algorithm works.
- The paper progressed smoothly from the background to the proposed algorithm.
- The layout of the text and plots are visually easy to digest.


**Weaknesses:**

- It wasn't too clear to me after reading the main text why having uniformly distributed credibility levels was the right thing to want.
- Similarly, it took a bit of drawing to see how all of the values in section 2 were related to each other.  Adding something like figure 9 to the main text could greatly help introduce the background material.


**Questions:**

- Why choose the one-sample Kolmogorov-Smirnov test to measure the divergence over others?

**Limitations:**

Yes, the authors acknowledge that their method is computationally expensive and may not scale to high dimensions as is.

---

> ### Author Rebuttal · Authors · 2023-08-08
>
> Dear Reviewer,
>
> Thank you very much for taking the time to review our manuscript and for your comments. Below we would like to address the questions:
>
> - *Why choose the one-sample Kolmogorov-Smirnov test to measure the divergence over others?*
>
>    The one-sample Kolmogorov-Smirnov test was our first choice. Its test statistic is straightforward to use as a minimization objective. The differentiable relaxation was also easy to find. We did not investigate alternative approaches.
>
>    As common alternatives for the KS test, the Anderson–Darling and Cramér–von Mises tests are typically listed. Both of them require ordered samples, thus only the sorting-based (alternatively, ranking-based) computation is available. Moreover, both of the tests rely on tabulated critical values which necessitates the introduction of significance level and is less convenient to use as a minimization objective.

---

> > ### Comment · Reviewer_T8qr · 2023-08-16
> >
> > Thanks, that makes sense.  I don't have anything other questions, the contributions are relevant and clearly presented in the paper.

---

### Author Rebuttal · Authors · 2023-08-08

Dear Reviewers,

Thank you very much for taking the time to review our manuscript and for your comments. Below we would like to address a question that has come up in several reviews:

*What are the advantages and disadvantages of using either the sorting-based computation or the direct computation method?*

First, we would like to underline that we made no claims of the superiority of the sorting-based computation (the one used in the experiments in the submitted manuscript) over the direct computation. Sorting-based computation happened to be the first one we evaluated empirically and since the method seemed to perform well we did not investigate the alternative approach in the light of limited computational resources.

Our recent empirical results - Figure 1 and Figure 2 in global response PDF - show that using direct computation instead of sorting-based computation while keeping all the remaining hyper-parameters untouched, leads to performance degradation both in terms of coverage and log-posterior density. $D_N$ is evaluated over 128 (same as the batch size) randomly sampled (in every iteration) levels on the (0,1) interval. We also observed optimization stability issues that reveal themselves in high variance over random initialization of the expected value of the approximate log posterior density of the nominal parameters for SpatialSIR in Figure 2 of the global response PDF.

We hypothesize that the poor performance of direct computation is due to the double use of a straight-through estimator through the indicator function - first in eg. (7) and then in eq. (8) - with the same backward relaxation. We see this issue as a direction for further investigation. Moreover, differentiable sorting can be seen as a straight-through estimator of a piecewise linear function suggesting an equivalence between sorting-based computation and direct computation but with different backward relaxations.

---

### Decision · Program_Chairs · 2023-09-21

**Decision:**

Accept (poster)

**Comment:**

This paper considers simulation-based-inference (SBI) in a setting where inference might be expected to return an inexact posterior. A novel regularizer is developed that encourages SBI to find well-calibrated posteriors despite being inexact. Reviewers agreed this was a useful and novel contribution.